# A Current Overview of the Use of Learning Analytics Dashboards

Italo Masiello [1],*, Zeynab (Artemis) Mohseni [1], Francis Palma [2], Susanna Nordmark [1], Hanna Augustsson [3] and Rebecka Rundquist [4]

1. Department of Computer Science and Media Technology, Linnaeus University, 352 52 Växjö, Sweden; zeynab.mohseni@lnu.se (Z.M.); susanna.nordmark@lnu.se (S.N.)
2. Faculty of Computer Science, University of New Brunswick, Fredericton, NB E3B 5A3, Canada; francis.palma@unb.ca
3. Department of Learning, Informatics, Management and Ethics, Karolinska Institutet, 171 77 Solna, Sweden; hanna.augustsson@ki.se
4. Department of Pedagogy and Learning, Linnaeus University, 352 52 Växjö, Sweden; rebecka.rundquist@lnu.se
* Correspondence: italo.masiello@lnu.se

**Abstract:** The promise of Learning Analytics Dashboards in education is to collect, analyze, and visualize data with the ultimate ambition of improving students' learning. Our overview of the latest systematic reviews on the topic shows a number of research trends: learning analytics research is growing rapidly; it brings to the front inequality and inclusiveness measures; it reveals an unclear path to data ownership and privacy; it provides predictions which are not clearly translated into pedagogical actions; and the possibility of self-regulated learning and game-based learning are not capitalized upon. However, as learning analytics research progresses, greater opportunities lie ahead, and a better integration between information science and learning sciences can bring added value of learning analytics dashboards in education.

**Keywords:** learning analytics dashboards; LAD; trends

## 1. Introduction

The promise of Learning Analytics (LA) in education is to collect and analyze data with the ultimate ambition of improving students' learning. Over the past decade, LA has transitioned from a mere concept to a solid area of study, encompassing a worldwide network of scholars and professionals. LA is defined as: "the measurement, collection, analysis, and reporting of data about learners and their contexts, for purposes of understanding and optimizing learning and the environments in which it occurs" [1] (pg. 252). Students and teachers interact with LA to process and consume educational data often through dashboards, also called Learning Analytics Dashboards (LADs). LADs are control panels that can be tailored to display LA components that update according to the learning processes, also in real time. Therefore, dashboards can serve as a medium employing LA to monitor, assess, and visualize learners' online behaviors related to learning, and other educational activities, such as study progress, grades, and group performance [2]. A subset of technical methods can be used for visualizing data in LADs. Examples of advanced LADs are presented in our latest research [3,4]. The visualizations mainly exemplify descriptive information, such as time spent on an online task, access to various online resources, and learning progression in a course or subject/task, but also provide means to compare user results. The dashboard visualizations of user data aim to increase pupil motivation, self-regulation, educational performance, and, furthermore, teacher engagement [5–7] on all education levels. With this potential, LADs are rapidly becoming integral assessment instruments for teaching and learning, and especially so in higher education, rather than at lower educational levels. This is because institutions within higher education often have more centralized data and technology infrastructure [8]. The COVID-19 pandemic functioned as a catalyst for an exponential growth in the use of digital technology in classrooms,

and the creation of data from this and its subsequent visualizations. Therefore, there is a clear need for the use of LADs in the classroom, especially if they can support teachers' pedagogical decision-making.

With the aim of understanding how far we have come and what the future reserves in this field, in this article, the authors synthesize and provide an overview of the latest systematic reviews examining the use of learning analytics and its dashboard visualization, identifying several trends and key areas for future research opportunities.

## 2. Methods

We started by checking if LA research was trending by simply conducting a search in Scopus for the term "learning analytics" from 2010 to 2023. This included 7496 publications related to LA. 'Title', 'Abstract', and 'Keywords' were considered as field tags. We examined the following document types: article, conference paper, conference review, book, book chapter, review, editorial, and data paper. With the knowledge that there was a large number of publications in this field, we then performed a so called 'umbrella review' [9], to compile evidence only from multiple systematic reviews, that is, literature bearing a high level of evidence, into a comprehensive article that allowed us to gain a broader perspective on the use of LADs in education and helped us identify trends.

We then followed the Joanna Briggs Institute's manual for evidence synthesis and followed the steps required in a systematic review of any evidence type [10], which includes: (1) formulating the research question, (2) identifying relevant studies and defining inclusion and exclusion criteria, (3) study selection (4), charting the data, and (5) collating, summarizing, and reporting the results.

### 2.1. Research Questions

With the aim of providing an overview of the latest systematic reviews examining the use of LADs, the main research question that guides this study was:

*What is the current use of learning analytics dashboards in education?*

The main research question is operationalized by a series of sub-questions designed to describe contextual and design aspects related to the use of LADs:

*Who are the users or target users? What are the visualization techniques? What design frameworks are used? What are the target outcomes? Are ethical issues discussed? And are strategies in play?*

Finally, one last sub-question is about understanding the evidence of the effects of LADs:

*What is the research evidence about the use of LADs?*

### 2.2. Identify the Relevant Studies

We searched for relevant systematic reviews in Scopus, Web of Science, and Scholar.Google.se. In those databases we searched for the terms: "systematic review" or "systematic literature review", and "learning analytics" in the title of articles, and "dashboard" in all fields. We also conducted a manual search on Google to complete the study's dataset, choosing all systematic reviews that met the criteria listed below.

The selection of papers was performed by the first and second authors and was based on the following criteria:

*Included studies*

1. That contained "learning analytics" and "systematic literature review" or "systematic review" in the title.
2. That contained "dashboard*" in the text.
3. That were published in English.
4. That were published only between 2019 and April 2023.

*Excluded studies*

1. That did not consider "learning analytics" as one subject or considered the two, each into a separate subject, that is, "learning" and "analytics".

### 2.3. Study Selection

The entire process of article selection returned 63 records published during the last 5 years, between 2019 and April 2023, in the three databases. Google search took into account all document types and year range, returning over 100,000 records. The two first authors then removed the duplicates and all articles that were outside the scope of this study. The majority of records in Google were also quickly removed since it returned the current systematic reviews within the first few screens or number of records. Lastly, 22 records were included in this study.

### 2.4. Data Charting

Table A1 in the Appendix A provides an overview of the included 22 systematic reviews about the use of LADs in education.

## 3. Reporting the Results

The final step is about reporting the results of the search process.

As we mentioned, before starting the review work, we engaged in a simple search exercise to see if the terms *learning analytics* was trending. The returned 7496 publications related to LA resulted in Figure 1. As shown in Figure 1a, over the past years, there has been a notable upward trend in publications related to LA. In 2010, only one publication was recorded in Scopus, and since then, there has been a consistent increase, reaching a peak of 983 publications in 2022. This growth reflected a growing interest in and recognition of the significance of LA, not just in education. The numbers indicated a nearly tenfold increase over the course of a decade, highlighting the expanding body of knowledge and research contributions in the field. The numbers of publications during the last five years further emphasized the sustained momentum and attention dedicated to LA in recent years. As can be seen in Figure 1b, research from the USA, Spain, Australia, the UK, Germany, and China represented the bulk of publication relevant to LA.

Table 1 illustrates a summary of the 22 reviews, considering the year of publication, country of study, the number of articles reviewed in the publication, and citations. As can be seen, Finland, with four articles, and the UK, USA, and Germany, with three articles each, have the most reviews on LADs. Moreover, the reviews published in 2019 and 2020 have already reached 193 and 290 citations, respectively.

### 3.1. Target Users

Most participants in the research analyzed in the reviews consisted of undergraduate students [2,11–21] and graduate students [2,11–13,19,20], or, if not specified, simply defined by 'higher education' students or "course demographics" [12,22–27], and teacher/lecturer group [16,21,27,28]. While virtually all reviews included studies conducted in higher education [2,12,13,16–19,26], studies with K12 students were fewer, and only two reviews focused solely on K12 [29,30]. This is comprehensible because of: (1) ethical issues; it is easier to ask an adult to participate in a research study where data security, privacy, and ethics are important implementation challenges [30,32]; (2) much of the educational data generated in higher education is made anonymous and freely available online with which to test visualization techniques [32]; (3) school systems are immensely diverse in terms of culture, customs, and attitudes toward technology and data [32]; and (4) schools occupy a politically sensitive place within society [32]. Indeed, this uncovers an opportunity to organize studies in this context.

### 3.2. Visualization Elements of Dashboards

Common visualization techniques were various graphs [2,12,13,19,21,23,25,26,28], such as line charts, bar charts, progress bars, pie charts, and timelines. In some of the reviews, there was no mention of what visualizations are represented in the dashboards [15–18,20,27,29,31]. Textual feedback is also an element used in dashboards [2,12,13,19,21,23,25,28], as well as tables [12–14,21,23]. According to the authors

of the reviews, visualization elements, when presented in meaningful ways, were fundamental to raise learners' and educators' awareness about learning processes in order to promote learning progression [13,21,23,26,31], as well as predict students' motivation, achievements, satisfaction, academic performance, and students at risk [11–14,19,21–23,25,26,31].

**Table 1.** Summary of the 22 systematic review articles, year, country of publication, number of articles reviewed, and citations (in ascending reference number order).

| Authors [Reference Number in this Paper] | Year | Country | # of Articles | Citations |
|---|---|---|---|---|
| Sahin & Ifenthaler [2] | 2021 | Germany | 76 | 19 |
| Sønderlund et al. [11] | 2019 | UK | 11 | 193 |
| Valle et al. [12] | 2021 | USA | 28 | 30 |
| Ahmad et al. [13] | 2022 | Germany | 161 | 8 |
| Avila et al. [14] | 2022 | Ecuador | 31 | 0 |
| Alwahaby et al. [15] | 2022 | UK | 100 | 28 |
| Tepgeç'& Ifenthaler [16] | 2022 | Germany | 52 | 2 |
| Elmoazen et al. [17] | 2023 | Finland | 21 | 14 |
| Khalil et al. [18] | 2023 | Norway | 26 | 6 |
| Moon et al. [19] | 2023 | USA | 27 | 3 |
| Heikkinen et al. [20] | 2023 | Finland | 56 | 13 |
| Ramaswami et al. [21] | 2023 | New Zealand | 15 | 5 |
| Cerratto Pargman & McGrath [22] | 2021 | Sweden | 21 | 57 |
| Matcha et al. [23] | 2020 | UK | 29 | 290 |
| Williamson & Kizilcec [24] | 2022 | USA | 47 | 34 |
| Banihashem et al. [25] | 2022 | Netherlands | 46 | 36 |
| Daoudi [26] | 2022 | Tunisia | 80 | 14 |
| Ouhaichi et al. [27] | 2023 | Sweden | 57 | 5 |
| Salas-Pilco et al. [28] | 2022 | China | 30 | 33 |
| Apiola et al. [29] | 2022 | Finland | 22 | 2 |
| Hirsto et al. [30] | 2022 | Finland | 33 | 3 |
| Kew & Tasir [31] | 2022 | Malaysia | 34 | 42 |

*3.3. Theoretical Framework Used in the Design of Dashboards*

Besides the obvious use of guidelines within the field of information science to support the appropriate design of visualization elements, it was also important to guide the design of LADs with relevant frameworks of pedagogy and learning to facilitate educational practices [20,21,26,33]. Among the theoretical frameworks, Self-Regulated Learning (SRL) [2,12–14,20,23,25,28] and game-based learning [13,17–19,23,26,30] theories seemed to be the most adopted for dashboard design. While a variety of less-used theories were motivation theory [2,13,23,25], open learner models [23], social learning theory [2], activity theory [12], inquiry-based learning [19], constructivism [12,29], collaborative knowledge construction [17], and multimodal learning or contextualizable learning analytics [15,19,27]. Human-centered design, i.e., functions and attributes of dashboards are defined by the people who are intended to use the system, and not imposed by designers and researchers, is also used [21,22,25,29]. Other reviews lacked any theoretical framework [14,16,23,24,31].

*3.4. Visualization Representations of Target Outcomes*

Dashboard visualizations in the various studies were used to track the following target outcomes: learner performance, progress, and competency level [2,11–13,15,17–21,23,25,26,28,29]; learning difficulties [13,18,23]; SRL [2,11–14,20,23,25,27,28]; awareness, reflection, and/or self-thinking [2,11–13,15,16,18,20,21,25–28,31]; affective measures such as motivation, anxiety, and satisfaction [2,11–13,15,16,23,25,26,31]; feedback practices [13–15, 21,25–27,29]; acceptance, such as ease of use and perceived usefulness [2,14,23]; behavioral measures such as time spent on activities and number of clicks; sequence of actions [2,11–13, 15,17,19,21,23,24,26,28,29,31]; knowledge creation [29]; and sensory measures [15,27]. There were also a number of other target outcomes, such as privacy, study skills, and learning

strategies [2,13,17,23]; group/collaborative learning [15,17,19,26,27,29]; and reward-based learning [23]. The target outcomes fell within the cognitive, behavioral, contextual, and affective domains.

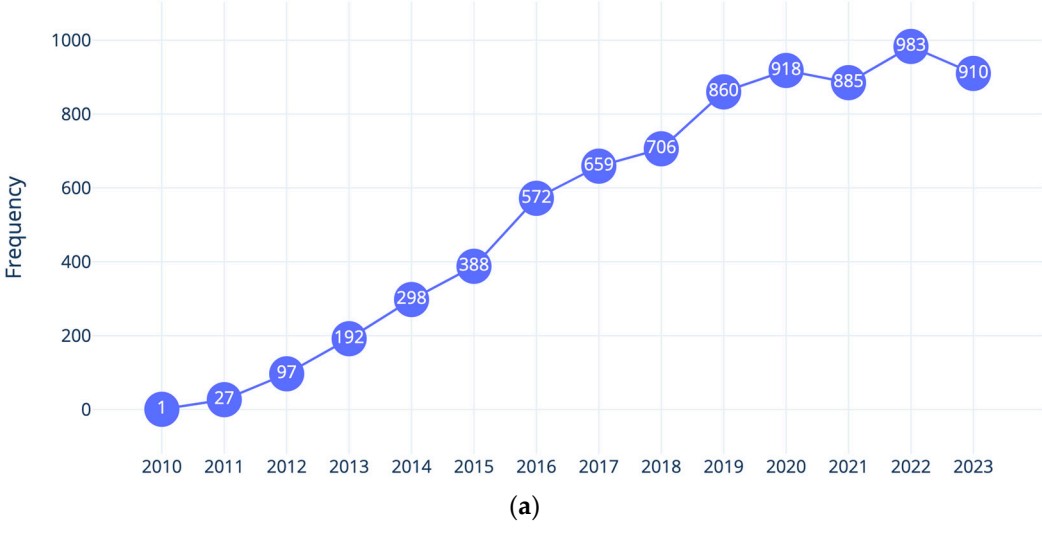

(**a**)

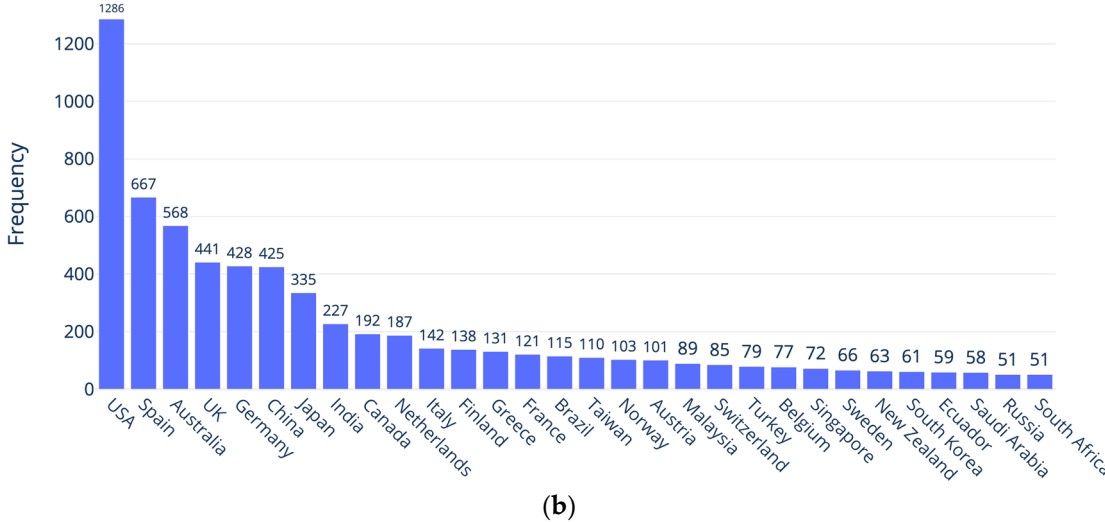

(**b**)

**Figure 1.** LA-related publications: (**a**) from 2010 to 2023; (**b**) in the first 30 countries with the highest number of publications.

### 3.5. Ethics of LA

As student data are increasingly being collected, processed, and visualized on dashboards, ethical issues are becoming prominent, making the use of student data a work in progress, both technically and pedagogically, as well as juridically. Among the ethics areas, privacy is a major concern [2,15,16,18,22,24–28,30,31] in relation to access, identification, and data governance of student data. Transparency is also a significant issue [15,22,28] in terms of source, analysis, and presentation of data. Other ethical issues are related to informed consent [15,22,28], i.e., raising awareness among students and teachers about the fact that their data is being tracked and analyzed; responsibility [22], that is, personal and institutional responsibility towards the use of student data; and validity [22,28] in relation to the accuracy of the algorithms used to interpret student data. However, despite the importance of the ethical issues, only a small percentage of LA research consider the relevance of ethics in this field [34].

### 3.6. Lacking Institutional Strategies

LA research has focused on the individuals, that is, the educator/teacher and student who are supposed to make sense of the analytics to contribute to teaching and learning, respectively [16]. Studies have shown that the responsibility to change learning behaviors when prompted by LA is put upon the higher education students, while to better reap the benefits of LA systemic interventions should be developed at the institutional level [11,14,18,21,22,31]. However, even though the legal, ethical, and effective utilization of LA is, indeed, the responsibility of the educational institution [15,21,22,30], a better interplay between the institution and students/teachers should foster better implementation, accountability [11,22], equity [24], and academic planning [23].

### 3.7. LAD Effectivenness

According to the reviews, the design of LADs should be guided by educational theories and pedagogical practices to be effective in helping educators and students to enhance teaching and learning [2,11–14,20–23,26]. The reviews also state that studies showing effectiveness in terms of reliably improved learning outcomes are steadily increasing yet few [15,20], corroborating earlier reviews [34], while the evidence of the effectiveness is growing in terms of motivation, engagement, and system usage behaviors [15,16,18,26–28,31]. One major obstacle is the methodologies of much of the research presented in the reviews, which relied on small convenience samples and simple correlational pre/post intervention designs. Ideally, and where possible, studies should also include experimental research methods like randomized control trials. Therefore, one way to increase effectiveness is by filling the gap between learning science and information science, that is, dashboard design led by learning and cognitive theories [31], and include systematic experimental research design [11], as earlier research also has pointed out, respectively [35,36].

## 4. Trends Based on the Research Reviews

The systematic review articles make an important contribution to a broader and general understanding of the affordances of LADs. Specifically, the authors can extract five trends in connection to the use of LADs that can be of value to practitioners and researchers, and ultimately learners. Those are presented below.

### 4.1. Trend 1. LADs Research Is Rapidly Evolving

The simple fact that only one review (within our timeframe) on this topic was published before 2020, while all others were published after (see Table 1), led us to believe that LADs' time is now. Perhaps one reason for this is that COVID-19 pandemic functioned as catalysator for an exponential use of educational technology in classrooms, and the exploitation of their generated data thereafter. This is in accordance with one of the reviews included here [2] and a recent article focusing on the challenges and limitations concerning LA [37]. This indicates that LADs research has been expanding rapidly, especially during the last few years, and the trend is expected to grow and evolve even more rapidly at all levels of education. This notion is also put forward in the yearly Horizon Report (Educause. https://www.educause.edu/; accessed on 4 December 2022) that annually identifies key educational technology trends. It has listed LA and its dashboard visualizations as an emerging technology. What this means is that more and more educational institutions are betting on the notion that LADs could help in determining several target outcomes based on data to help the appropriate decision-making process in regard to students and teachers, and teaching and learning. Yet, there are only limited examples of large-scale and well-developed LADs implementations across educational institutions [38]. This leaves a door wide open for developers to capitalize on this trend.

### 4.2. Trend 2. Inequalities and Inclusiveness

The papers on LA are published in educational technology, education, and engineering journals [2]. As can be seen from Figure 1b, the origins of the papers are heavily concen-

trated in North America, Europe, and Australia, and Asian countries also have a large footprint. This means that the development of LADs and its research is carried out in specific contexts. In a long-term perspective, this can become a bias. In fact, what and how data is visualized, is influenced by software developers and the specific context into which the dashboards—through various algorithms—are developed. This limits the broader implementation in other countries without a thorough local evaluation, which could result in problems, such as embedded racial or gender stereotypes, or unjust predictions [24,39]. This could be amplified by the fact that LA research is lacking institutional strategies, exemplified in many of the reviews included here. Also, background information about learners like demographic data (gender, sex, age, race etc.) is not often reported in the included studies [15,24,26]. Moreover, broader aspects of inclusiveness and disability are seldom researched [18,23]. In addition, governments are lagging behind in comprehending the use of analytics, for example in K12 education [32]. Verbert et al. [7] state that it is necessary to address the implications of predictive systems and bias in LA research, and making the visualizations transparent can offer a better understanding of the decision-making model for the users. This trend run contrary to the United Nations' Sustainable Development Goal 4–Quality Education–which aims to "ensure inclusive and equitable quality education and promote lifelong learning opportunities for all" (https://sdgs.un.org/goals/goal4; accessed on 24 October 2023).

### 4.3. Trend 3. Data Privacy, Ownership, and Legal Framework

The challenges of data privacy and ownership are, according to the reviews [2,15,16, 18,22,24–28,30,31], not sufficiently researched and discussed, as also pointed out in recent research literature [37,40]. However, those are issues that could ultimately end up being politicized, possibly discouraging further development of LADs, if not properly attended at the right level, i.e., between students or legal guardians, teachers, and the respective educational institutions [15,31]. Legal frameworks are proposed to mitigate concerns over LA [37,40], although this is a relatively new phenomenon with few examples in practice, which leaves room for interpretation of legal frameworks in different contexts [39]. A key ethical concern is data ownership. "Those who own the data own the future", writes Harari [41] (pg 89). According to the author, data ownership is going to bring even further inequalities into the world [41]. This and the previous trend seem to be interrelated. The tech giants like Facebook, Google, and Amazon are amassing profits based on our personal data use, and they feed us with information based on that data to capture our attention. Those same tech giants are also part of the education domain. What stops them from steering our learning behaviors for their own profit, monetary or otherwise? Here, we argue that each of us should own our own data, period!

### 4.4. Trend 4. Prediction and Effectiveness—Lost in Translation

The reviews are in agreement about the notion that LA seems to provide efficient predictive models [11–13,15,18,21,22,25], as research has already indicated [37,40]. Prediction algorithms are highly specialized and can predict retention [11,13,18,20,21,25,31,34], study success, test score, drop-out, and students' well-being behavior [11,13,25]. Those predictions are turned into visualization elements to raise teachers' and students' awareness so that they can act upon said elements. Therefore, the trend is to design dashboards to be more effective and efficient by connecting learning theories to dashboard design [2,12,13,20,21,23]. However, that in itself is not sufficient for either teachers or students to enhance the effectiveness of dashboards and visualizations elements, or as Sønderlund et al. [11] put it: "how this knowledge is translated into interventions". What this means is that a LAD in itself is not a necessary condition to bring behavioral changes to educational processes that can be solved using traditional means. As with all educational technology, regardless of type or purpose, we need to devise a method for translating the possible effect into practice. According to the reviews, the research evidence of their effectiveness in improving teaching or learning is growing [11,16,25,26], yet still limited [11,15,20,21]. Indeed, there is a scarcity

of credible studies in the peer-reviewed LA literature that demonstrate significant impact on learning or student success [38].

*4.5. Trend 5. Self-Regulated Learning and Game-Based Learning*

SRL refers to learning guided by intrinsic or extrinsic motivation to learn and to plan, monitor, and assess personal progress and understanding to adjust for the next learning task [42]. Game-based learning refers to learning and instruction through the use of a digital application that combines aspects of video games, [43], gamification, and has the potential to enhance learners' knowledge gain and comprehension, motivation, and performance [26].

There are several SRL models proposed in the literature [44–49], and many have been incorporated into MOOCs (Multiple Online Open Courses), or online courses with a self-paced structure, like those exemplified in the reviews [2,12,14,20,23,27]. A similar situation is reached with game-based learning, that is, gamification concepts are introduced in many classroom and online learning situations and users of diverse demographics [15,17,18,23,26,30]. In the articles included in the reviews, the authors visualize students' progress, i.e., course or module completion, by using progress bars or charts. Unfortunately, visualization per se is not sufficient for effective feedback either for self-regulation or game-based learning [23,26]. SRL guiding rules, i.e., metacognition, motivation, and emotion, also need to be accommodated into LADs [23], as well as demographic data and previous game experience [26]. The reviews have highlighted that existing LADs are rarely grounded in learning theories [23,24]. In this study though, we have demonstrated that there are various frameworks to guide the design of LADs. In addition, there is an immense opportunity to incorporate SRL and game-based learning into LADs. SRL provides an opportunity, especially in higher education, where online, self-paced learning is growing, and students have recognized a possible value of LADs suited for SRL [22]. Furthermore, game-based learning could provide an opportunity to support a whole range of intellectual disabilities in lower grades [18].

**5. Conclusions**

The latest systematic review articles on LA and dashboards have different focuses, therefore there is a variety of contextual and design aspects in connection to LADs. The reviews report the bulk of the use of LA within higher education, and the use of simple statistical graphs as visualization elements. The reviews also address the use of educational theories to guide the design of LA dashboards, with SRL and game-based learning being prevalent theories, and the tracking of outcomes such as users' performance, behaviors, and effective measures. Furthermore, the reviews note a raising awareness of ethical issues aligned with the growing use of LADs, and the lack of organizational policies to make the use of LA more efficient, as well as accountable and equitable. Last, but not least, the reviews reveal a sparsity of studies that show effectiveness of improved learning outcomes—the promise of LA—even though the evidence is growing.

Based on the research evidence exemplified in the review articles, the researchers have identified five emerging trends in connection to LADs. The first and most obvious one is that LA and its visualization through dashboards is being increasingly used, especially in higher education and western countries, but also growing at lower educational levels and Asian countries. This means that LADs afford individuals or organizations to train educational decision-making in a hybrid format, where the 'human' is aided by the 'machine'. The second emerging trend concerns the fact that the development of LA and dashboards occurs in specific regions of the world, often omitting demographics and disability aspects, which projects development inequalities and exclusiveness in the LA algorithms used in LADs. The third trend highlights the ethical and legal complexity related to the analytics of educational data, pinpointing the absolute necessity for a deeper discourse with all stakeholders, and a clearer way forward about a LA legal and ethical framework on all levels of education. The fourth trend is all about prediction, and little about learning

effectiveness. LA algorithms are effective at predicting. However, the disconnection to practical pedagogical interventions limits the educational value of LA. The final and fifth emerging trend argues about the potential of LA built into LADs to support SRL and game-based learning. However, there is a need for a clearer connection between the design of LADs and what educational science asserts works for learning.

We are aware that our overview is limited by the fact that we only chose to look at systematic reviews of different quality and available data, raising the issue of heterogeneity when combining reviews with different conditions. Nonetheless, the authors believe that this overview is useful in giving an intuitive analysis of this emerging field of research. As time and development of LADs progresses, we will discover new trends. However, the authors believe that researchers and developers need to team up to collaboratively understand better the processes involved in learning and teaching practices and bring them into the design of LADs to expand the possibilities and value of analytics for practitioners in education [50,51]. We offer a number of key areas for future research opportunities at the intersection of the fields of LA and education science:

- Establish an international research agenda to test and develop LA and LADs in a cross-cultural and cross-language manner to maximize possible benefits.
- Move the evaluation of LADs beyond functionality and usability aspects and assess the impact on the usefulness of LADs to increase understanding not only on outcomes but also learning processes.
- Examine predictive analytics of LADs and the self-reflection that they elicit in learners which results in positive behavioral and cognitive adjustments, incorporate learning sciences into the design of LADs, and use systematic experimental research methods for learning at scale.
- Examine and address possible biases towards different user groups and demographics in the design of LA and LADs.
- Consider ethical aspects of educational and personal data protection such as privacy, security, and transparency. Develop national and international legislation methods for data collection and analysis.

The promise of LA and respective LADs seems to be "alive and kicking".

**Author Contributions:** Conceptualization, I.M.; methodology, I.M. and Z.M.; validation, I.M. and Z.M.; formal analysis, I.M., Z.M. and F.P.; data curation, I.M. and Z.M.; writing original draft presentation, I.M.; writing, review and editing, I.M., Z.M., F.P., S.N., H.A. and R.R.; visualizations, Z.M.; funding acquisition, I.M. All authors have read and agreed to the published version of the manuscript.

**Funding:** This work was supported by the Swedish Research Council for Health, Working Life and Welfare, grant number Forte 2020-01221 and Växjö Kommun, grant number 2020/3209-5.1.1.

**Institutional Review Board Statement:** Not applicable.

**Informed Consent Statement:** Not applicable.

**Data Availability Statement:** No new data were created.

**Conflicts of Interest:** The authors declare no conflicts of interest.

## Abbreviations

| | |
|---|---|
| LA | Learning Analytics |
| LAD | Learning Analytics Dashboards |
| SRL | Self-Regulated Learning |

## Appendix A

**Table A1.** An overview of the review articles on LADs. The first column reports the publication with respective authors and year [the reference number in this paper]. The second column reports the summary of the publication.

| Study—[Reference Number] Authors, Year and Title | Summary |
|---|---|
| [2] Sahin, M., & Ifenthaler, D. (2021). Visualizations and Dashboards for Learning Analytics: A Systematic Literature Review. | Performed a descriptive systematic analysis, fitting their criteria (i.e., containing the search string 'Learning Analytics', and 'Learning Analytics' and 'Dashboard' or 'Visualization'). Those articles were divided into eight summative and descriptive categories: keywords, stakeholders (target group) and year, study group (participants), visualization techniques, methods, data collection tools, variables, and theoretical background. The highest number of articles (six) that included the search string were found in the proceedings of the International Conference on Learning Analytics and Knowledge and in the Computers & Education journal (four), with a sharp increase of the number of publications from 2017 to 2021. |
| [11] Sønderlund, A.L., Hughes, E., & Smith, J. (2019). The efficacy of learning analytics interventions in higher education: A systematic review. | Synthesized the research on the effectiveness of LA intervention on higher education students' underachievement, experience, and drop-out. The authors reviewed 11 articles from the USA, Brazil, Taiwan, and South Korea. The authors also compare past and current LA methods. |
| [12] Valle, N., Antonenko, P., Dawson, K., & Huggins-Manley, A. C. (2021). Staying on target: A systematic literature review on learner-facing learning analytics dashboards. | Studied the design of LA dashboards, the educational context where the dashboards are implemented, and the types and features of the studies in which the dashboards are used. Twenty-eight articles were included, with more than half published in scientific journals, 36% in proceedings, and the remaining were dissertations, all based in western countries. The authors debated how affects and motivation of users of LA dashboards have been disregarded in the research on the use of LA dashboards. |
| [13] Ahmad, A., Schneider, J., Griffiths, D., Biedermann, D., Schiffner, D., Greller, W., & Drachsler, H. (2022). Connecting the dots–A literature review on learning analytics indicators from a learning design perspective. | Investigated the alignment between learning design (LD) and LA. The review analyzed 161 LA articles to identify indicators based on learning design events and their associated metrics. The proposed reference framework in this review aimed to bridge the gap between these two fields. The study identified four distinct ways in which learning activities have been described in LA literature: procedural actions, LD activities, a combination of LD activities and procedural actions, and scenarios where no explicit activities were mentioned. Furthermore, 135 LA indicators were categorized into 19 clusters based on their similarities and goals, with "predictive analytics", "performance", and "self-regulation" being the most prevalent clusters. It also discussed the importance of aligning LA with pedagogical models to improve educational outcomes and the need for clear guidelines in this regard. |
| [14] Avila, A. G. N., Feraud, I. F. S., Solano-Quinde, L. D., Zuniga-Prieto, M., Echeverria, V., & De Laet, T. (2022, October). Learning Analytics to Support the Provision of Feedback in Higher Education: a Systematic Literature Review. | Investigated the use of LA feedback tools to enhance SRL skills in higher education. The review covered articles published over the past 10 years, resulting in the analysis of 31 papers. While LA feedback tools are considered a promising approach for improving SRL skills, the majority of the reviewed papers lack a strong theoretical basis in SRL. This review highlighted the need for further empirical research with a focus on both quantitative and qualitative approaches, encompassing a wider range of educational settings and outcome measures to better understand the impact of LA feedback tools on SRL skills, well-being, and academic performance. |
| [15] Alwahaby, H., Cukurova, M., Papamitsiou, Z., & Giannakos, M. (2022). The evidence of impact and ethical considerations of Multimodal Learning Analytics: A Systematic Literature Review. | Reviewed 100 articles published between 2010 and 2020, focusing on Multimodal Learning Analytics (MMLA) research. This review aimed to address research questions related to the real-world impact on learning outcomes and ethical considerations in MMLA research. The papers were coded based on data modalities and types of empirical evidence provided, including causal evidence, correlational evidence, descriptive evidence, anecdotal evidence, prototypes with no evidence, and machine learning. |

**Table A1.** *Cont.*

| Study—[Reference Number] Authors, Year and Title | Summary |
|---|---|
| [16] Tepgeç, M., & Ifenthaler, D. (2022). Learning Analytics Based Interventions: A Systematic Review of Experimental Studies. | Focused on the field of LA and its impact on educational interventions. It highlighted the growing interest in LA but noted the lack of empirical evidence on the effects of such interventions. The review included 52 papers. The findings indicated that student-facing dashboards are the most commonly employed LA-based intervention. Additionally, the review discussed the methodological aspects of these interventions, noting that data distillation for human judgment is the most prevalent approach. In terms of outcomes, the review looked at learning outcomes, user reflections, motivation, engagement, system usage behaviors, and teachers' performance monitoring. |
| [17] Elmoazern, R. Saqr, M., Khalil, M., & Wasson, B. (2023). Learning analytics in virtual laboratories: A systematic literature review of empirical research. | Examined the current research on LA and collaboration in online laboratory environments. A total of 21 articles between 2015 and 2021 were included. Half of the studies were within the higher education and medical field, but also lower level of education. There were a broad variety of lab platforms and LA was derived from students' log files. LA was used to measure performance, activities, perceptions, and behavior of students in virtual labs. The landscape was fragmented, and the research did not focus on learning and teaching. Standards and protocols are needed to address factors of collaboration, learning analytics, and online labs. |
| [18] Khalil, M., Slade, S., & Prinsloo, P. (2023). Learning analytics in support of inclusiveness and disable students: a systematic review. | Examined the role of LA in promoting inclusiveness for students with disabilities. A total of 26 articles were analyzed, and the results indicated that while LA began in 2011, discussions on inclusiveness in education started only in 2016. LA has the potential to foster inclusiveness by reducing discrimination and supporting marginalized groups. However, there are gaps in realizing this potential. |
| [19] Moon, J., Lee, D., Choi, G.W., Seo, J., Do, J., & Lim T. (2023). Learning analytics in seamless learning environments: a systematic review. | Reviewed the use of LA within seamless learning environments, analyzing 27 journal articles. While seamless learning, which focuses on inquiry-based and experiential learning, has grown in popularity, the integration of LA to assess student progress in these environments has also increased. However, there's a gap in comprehensive reviews on this integration. |
| [20] Heikkinen, S., Saqr, M., Malmberg, J., & Tedre, M. (2023). Supporting self-regulated learning with learning analytics interventions—a systematic literature review. | Analyzed studies that employed LA interventions to boost SRL and found 56 which met the criteria. While various LA interventions aimed to support SRL, only 46% showed a positive impact on learning, and just four studies covered all SRL phases. The findings suggest that interventions should consider all SRL phases and call for more comparative research to determine the most effective strategies. |
| [21] Ramaswani, G., Susnjak, T., Mathrani, A., & Umer, R. (2023). Use of predisctive analytics with learning analytics dashboards: A review of case studies. | Analyzed 15 studies revealing that current LADs lack advanced predictive analytics, mostly identifying at-risk students without providing interpretative or actionable advice. Many LADs are still prototypes, and evaluations focus on functionality rather than educational impact. The study recommends creating more advanced LADs using machine learning and stresses the need for evaluations based on educational effectiveness. |
| [22] Cerratto Pargman, T., & McGrath, C. (2021). Mapping the Ethics of Learning Analytics in Higher Education: A Systematic Literature Review of Empirical Research. [23] Matcha, W., Uzir, N. A., Gašević, D., & Pardo, A. (2020). A Systematic Review of Empirical Studies on Learning Analytics Dashboards: A Self-Regulated Learning Perspective. | Presented the latest evidence on the ethical aspects in relation to the use of LA in higher education. They reviewed 21 publications (16 journal articles and five proceedings), with a concentration of the research mostly in a few countries: the USA, Australia, and the UK. Those aspects concerned mainly the transparency, privacy, and informed consent that if not properly attended, according to the evidence, can prevent the development and implementation of LA.<br><br>Reviewed studies that empirically assess the impact of LADs on learning and teaching based on a SRL model. The review included 29 papers published between 2010 and 2017. Based on the review, the authors proposed a user-centered learning analytics system (MULAS). MULAS should be used to guide researchers and practitioners in understanding and developing learning environments, instead of making any design decisions on representation of data and analytics results. |

**Table A1.** *Cont.*

| Study—[Reference Number] Authors, Year and Title | Summary |
| --- | --- |
| [24] Williamson, K., & Kizilcec, R. (2022). A Review of Learning Analytics Dashboard Research in Higher Education: Implications for Justice, Equity, Diversity, and Inclusion. | Discussed issues such as justice, equity, diversity, and inclusion in relation to LA dashboards research in higher education. The review included 45 relevant publications from journals and two conference proceedings, mostly from North America, Europe, and Australia. The authors identified four themes: participant identities and researcher positionality, surveillance concerns, implicit pedagogies, and software development resources. All were needed to avoid the risk of reinforcing inequities in education, according to the authors. The review also considered how to address those issues in future LA dashboards research and development. |
| [25] Banihashem, S. K., Noroozi, O., van Ginkel, S., Macfadyen, L. P., & Biemans, H. J. (2022). A systematic review of the role of learning analytics in enhancing feedback practices in higher education. | The study reviewed the status of LA-based feedback systems in higher education. The review included 46 papers published between 2011 and 2022 and presented the classification of LA into four dimensions: (1) what (what types of data does the system capture and analyze); (2) how (how does the system perform analytics); (3) why (for what reasons does the system gather and analyze data); and (4) who (who is served by the analytics). The 46 publications are mostly from Australia, European, and Asian countries, while the USA, south America, Canada, and African countries have one. |
| [26] Daoudi, I. (2022). Learning analytics for enhancing the usability of serious games in formal education: A systematic literature review and research agenda. | Explored the use of Serious Educational Games (SEGs) and the application of Game Learning Analytics (GLA) in formal education settings. The review included 80 relevant studies, and the key findings were: (1) GLA emerged as a powerful tool to assess and improve the usability of educational games. It has the potential to enhance education by improving learning outcomes, early detection of students at risk, increasing learner engagement, providing real-time feedback, and personalizing the learning experience). (2) Multidimensional taxonomy to categorize and understand different aspects of SEGs in formal education. |
| [27] Ouhaichi, H., Spikol, D., & Vogel, B. (2023). Research trends in multimodal learning analytics: A systematic mapping study. | Identified research types, methodologies, and trends in the area of MMLA, revealing an increasing interest in these technologies. The authors reviewed 57 studies and reviews and highlighted 14 topics under four themes—learning context, learning processes, systems and modality, and technologies—that can contribute to the development of MMLA. |
| [28] Salas-Pilco, S. Z., Xiao, K., & Hu, X. (2023). Correction to: Artificial intelligence and learning analytics in teacher education: A systematic review | Discussed the integration of artificial intelligence (AI) and LA in teacher education. They analyzed 30 studies. Key findings from this review reveal a concentration on examining the behaviors, perceptions, and digital competencies of pre-service and in-service teachers regarding AI and LA in their teaching practices. The primary types of data utilized in these studies include behavioral data, discourse data, and statistical data. Furthermore, machine learning algorithms are a common feature in most of these studies. However, only a few studies mention ethical considerations. |
| [29] Apiola, M. V., Lipponen, S., Seitamaa, A., Korhonen, T., & Hakkarainen, K. (2022, July). Learning Analytics for Knowledge Creation and Inventing in K-12: A Systematic Review. | Reviewed empirical LA studies conducted in K-12 education with a focus on pedagogically innovative approaches involving technology-mediated learning, such as knowledge building and maker-centered learning. The review identified 22 articles with an emphasis on constructivist pedagogies. A majority (eight) of the included articles took place in the secondary school (grades 7–9, age 13–15). Four articles took place in primary school (grade 1–6) and four in high school. Another four articles had a variety of age groups ranging between 10 and 18 years. Finally, two articles had teachers or pre-service teacher participants. |
| [30] Hirsto, L., Saqr, M., López-Pernas, S., & Valtonen, T. (2022). A systematic narrative review of learning analytics research in K-12 and schools. | Analyzed the impact of LA research in the context of elementary level teaching. The review focused on highly cited articles retrieved from the Scopus database, with a total of 33 papers meeting the criteria. The analysis revealed that LA research in elementary education is relatively limited, especially in terms of highly cited studies. The research within this field is fragmented, with varying pedagogical goals and approaches. It was noted that LA is used both as a means for analyzing research data and as a tool to support students' learning. The review emphasized the importance of developing more robust theoretical foundations for LA in elementary school contexts. |

**Table A1.** *Cont.*

| Study—[Reference Number] Authors, Year and Title | Summary |
| --- | --- |
| [31] Kew, S. N., & Tasir, Z. (2022). Learning analytics in online learning environment: A systematic review on the focuses and the types of student-related analytics data. | Presented a comprehensive overview of LA in online learning environments, drawing from a review of 34 articles published between 2012 and 2020. The review highlighted the key themes in LA research, such as monitoring/analysis, prediction/intervention, and the types of student-related analytics data commonly used. Additionally, it addressed challenges in LA, including data management, dissemination of results, and ethical considerations. |

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
