# Peer review of "A Current Overview of the Use of Learning Analytics Dashboards"

_education, doi:10.3390/educsci14010082_

Round 1

Reviewer 1 Report

Comments and Suggestions for Authors

The authors conducted a meta-review to discover themes and trends regarding learning analytics dashboards. 21 key studies were identified and analyzed to draw conclusions on the themes that SLR from 2019 till today focuses on. To extract current trends within the literature, two more searches were conducted. The combined results lead to the formulation of five trends. Finally, based on their synthesis of the extant literature, the authors state five explicit directions for future research.

The paper itself is well-written, and the line of thought is easy to follow. It provides a neat overview of recent systematic reviews and a nice summary of the topics that are discussed in these reviews. Nonetheless, the paper has some serious shortcomings. First and foremost: the research methodology is not sound. It looks like some form of ‘convenience sampling’ took place, where some interesting articles from Scopus and Scholar were used in the analysis. When I use the same search terms in Scholar, I find articles that are not incorporated in the study at hand. So, what inclusion and exclusion criteria were used? The authors write that they did not conduct a comprehensive systematic review (row 46), but why didn’t you? Also, there is no information on the analysis phase of the research. Who were involved in the analysis, and how were articles analyzed? How did the themes emerge from the key studies, and how do you define a “theme”? Some of them relate to design principles (visuals, theoretical frameworks), while others relate to contextual factors (educational level, institutional strategy). Later, in the section about trends, all the sudden two more searches are conducted to collect additional information. This does not strike me as scientifically sound. Again, how do you define "trend"? The paper also lacks a specific research question. There is a question (“How far have we come and what does the future reserve in this field”) but that is too vague for me.

The title is somewhat misleading. That is, the paper does not measure or describe the impact of learning analytics dashboards on learning. LAD effectiveness is one of the identified themes but not the article’s main focus. Moreover, the inclusion of the term EdTech strikes me as odd. It is mentioned a couple of times but learning analytics is not positioned within the context of EdTech. You looked at trends within learning analytics literature regarding dashboards, but in the conclusion it is stated that “the researchers have identified five emerging EdTech trends in connection to LADs” (rows 290-291). I do not see the connection between your research and EdTech trends in general.

Figure 2 shows a word cloud containing country names. It is unclear what these countries stand for: do they represent the location where data was collected, or where authors are coming from? I also believe that this form of visualization is not very suitable for the goal you try to achieve. It is very hard to draw detailed conclusions from it and come to the same conclusions. Why not a bar chart with the number of articles per continent? Moreover, the authors state that “the development of LADs and its research is carried out in specific contexts” (rows 200-201). Nonetheless, many countries and four out of six continents are clearly present. Also mind the figure’s caption: it says 2017-2023 but the text says 2020-2023. Which one is true?

Overall, I like the paper as it provides a clear yet uncomplete overview of recent reviews regarding LADs. However, I believe that the paper has some major shortcomings which makes it unsuitable for publication in this journal.

Comments on the Quality of English Language

Overall, the quality of English is good. Some sentences seem a bit odd, and I am not sure whether all the prepositions are correct. However, for me as a non-native English speaker, it is hard to make definite statements about this. Nonetheless, proofreading by a native English speaker never hurts.

Author Response

We greatly appreciate the comments that the reviewers presented. We recognize the issue of not presenting a better section of the Methods and the possible confusion that this have brought. We honestly try to keep the paper as short as possible and avoid complicated lengthy description and analysis, even though we ran a very thorough analysis from the beginning. That is why we avoided the more systematic work. But we realized that that was counterproductive. Therefore, we recompiled our initial review work and this time followed the systematic analysis work of Arksey and O’Malley, as explained on page 2.

We also added a Method section and explain the process of review according to the comments from the reviewers. We included the details of the search, research questions, inclusion criteria, and study selection on page 2.

If some articles from Google Scholar are missing in our paper it is because it either was not a systematic review or did not fit the timeframe we used, that 2019 – April 2023. But we added one systematic review that we missed the first time, nr 28.

We also explain what the purpose of the two searches was on page 2 and the results of that on page 3.

As the reviewer rightly suggest, we looked at design principles and contextual factors. We have explained this under research questions on page 2. Thanks for the suggestion!

We changed the title to better reflects the focus of the article, that is, the use of LADs.

We have also removed the term EdTech, as suggested by the reviewer.

We have removed Figure 2 and replace it with Figure 1 (a-b) on page 4. This is a simple bar chart and a line graph that are easy to read and understand.

Reviewer 2 Report

Comments and Suggestions for Authors

The topic of the article is interesting and timely. The authors have attempted to conduct a review of other reviews regarding the use of Learning Analytics Dashboards.

The article lacks a coherent and clear methodology for conducting the review. Several issues support this assessment:

*  In the introduction (Section 2), it is stated that two databases, Scopus and Google Scholar are utilized. However, later in the text (Section 4.1), it is mentioned that only Google Scholar is employed for another segment of the study. This lack of consistency is noteworthy. I recommend considering the use of Scopus and Web of Science for such studies, as they index high-quality and peer-reviewed material.

* The searches conducted with the queries in Section 2 are not executed effectively, and the results are not sufficiently processed. The terms employed lead to the exclusion of results; for instance, the use of only 'systematic review' is limiting, and additional terms should be incorporated, such as ‘Literature review’ or ‘Research review’. Additionally, in the section they wrote "hand search," using very few terms and mentioned an "...arbitrary chose...," which lacks clarity.

* The authors do not provide indications or details about the process employed for conducting the study. I recommend adopting a standardized methodology, such as the PRISMA 2020 protocol, to enhance transparency and rigor in the study design.     

* The authors do not specifically indicate in which parts of the articles they search for their terms (title, abstract, keywords, or other sections). They also do not clarify the type of reading they undertake on the articles—whether they only read certain sections or the entire article. Furthermore, there is no mention of whether they employ inclusion or exclusion criteria for the articles to be analyzed in their review.

All these deficiencies could account for the exclusion of studies that should have been analyzed. At the very least, I would suggest including the following review in the search and analysis, which I recommend the authors consider: Martínez Monés, A., Dimitriadis Damoulis, Y., et al. Achievements and challenges in learning analytics in Spain: The view of SNOLA. RIED Revista Iberoamericana de Educación a Distancia. 2020;23(2):187. https://doi.org/10.5944/ried.23.2.26541

The authors present some study findings by indicating the references that correspond to each statement. However, the article's information is excessively fragmented, as demonstrated in this example, making it challenging to read:  “… competency level [2,9–11,13,15–108 19,21,23,24,27], learning difficulties [11,16,21], SRL [2,9–12,18,21,23,25], awareness, 109 reflection, and/or self-thinking [2,9–11,13,14,16,18,19,23–25,29], affective measures, such 110 as motivation, anxiety, satisfaction [2,9–11,13,14,21,23,24,29], feedback practices [11–111 13,19,23–25,27] …”

I recommend that the authors synthesize the results in tables, presenting numerical values or percentages to enhance the readability of the findings.

 Figure 1 is challenging to comprehend, and Figure 2 appears to lack relevance. I encourage the authors to enhance both figures, opting for a different type of representation that effectively communicates the intended information.

The classification of articles by educational level and the exploration of dashboard elements are interesting. However, it might be beneficial for the authors to conduct a comprehensive analysis of these topics to determine, for instance, if a singular type of dashboard is predominantly used in higher education. A more in-depth analysis is warranted.

Author Response

We greatly appreciate the comments that the reviewers presented. We recognize the issue of not presenting a better section of the Methods and the possible confusion that this have brought. We honestly try to keep the paper as short as possible and avoid complicated lengthy description and analysis, even though we ran a very thorough analysis from the beginning. That is why we avoided the more systematic work. But we realized that that was counterproductive. Therefore, we recompiled our initial review work and this time followed the systematic analysis work of Arksey and O’Malley, as explained on page 2. We did  follow the comments and used Scopus, Web of Science and Scholar.

We used the term “systematic review” or “systematic literature review”, explained on page 2. Either of the terms that should be used in a title, according to the prominent Cochrane’s reporting of literature reviews, if a systematic review is performed. We also explained on page 2 the differences with the two searches we have done. We hope it is clearer.

We did not use Prisma but the Arksey and O’Malley methods. And since we did not find many records we did not believe we needed to be more specific. Even though Google search gave over 100,000 hits, those were easily managed in a few minutes, as only the first few 30-40 were related to our focus.

On page 2 we have specified how the search was done and which field we searched.

We followed the suggestion of the first reviewer and described that we looked at design principles and contextual factors. We have explained this under research questions on page 2. We read the entire articles.

We defined the inclusion and exclusion criteria on page 2.

The paper by Martinez-Mones et al. that the reviewer suggested does not seem to be a systematic review. We realize that looking only at systematic reviews is limiting, but we wanted to use the data from systematic reviews which bears a high level of evidence, as explained on page 2. We explain this as a limitation in the conclusions.

We understand that presenting the references in the text may look fragmented, but we think this is anyway the simplest way to do it in order to keep the text of the manuscript as short as possible. We did not think that presenting the results in table makes it easier to read.

We have changed the Figures and adjusted to fit the comments of the reviewers.

We believe we did perform an in-depth analysis without being excessively lengthy so that we can get greater readership.

Reviewer 3 Report

Comments and Suggestions for Authors

1. Does the article address a topic of current concern in relation to the aims of the journal?

Yes. This paper reviews the latest studies on learning analytics dashboards that play a role in digital teaching and learning.

2. Does the abstract give a clear account of the scope of the paper?

Yes. The abstract highlights the key points of the paper.

3. Do the keywords adequately reflect the paper?

Yes.

4. Does the article consider relevant contemporary literature in the area?

Recent related studies are reviewed.

5. If the article is concerned with research activity, has a sound methodology been employed and described?

Some suggestions to better this article, as described below.

-       The mechanism of paper filtering, selection, and reviewing should be described in the method section.

-       Several trends were concluded according to authors’ comprehension and interpretation, and it would be great if other raters’ perspectives or the coding & clustering methodology can be added to enhance the validity and reliability.

-       Please provide some classic screenshots of LADs for readers.

6. Does the article clearly distinguish between opinion and empirical evidence?

Yes.

7. Does the article contribute to a critical understanding of the issues?

It may provide e-learning and ed-tech researchers and practitioners with a better understanding of LAD trends and related issues.

8. Does the article clearly reference citations and quotations according to the submission guidelines?

Yes.

Author Response

We have added a Method section and explained in more details the review process and the coding.

In the final version we lead the reader towards our published LADs, so that the reader can get an understanding. Choosing one single LAD to show in this paper would not do justice to the myriads of LADs out there.

Round 2

Reviewer 1 Report

Comments and Suggestions for Authors

The authors did an excellent job improving the paper. The first version was below quite below the standards for a research paper but the new version is much better. My concerns about the research methodology are well addressed as the authors now provide a good explanation about their research approach. I suggest to accept the paper for publication.